# REPRESENTATION QUALITY EXPLAINS ADVERSARIAL ATTACKS

## ABSTRACT

Neural networks have been shown vulnerable to adversarial samples. Slightly perturbed input images are able to change the classification of accurate models, showing that the representation learned is not as good as previously thought. To aid the development of better neural networks, it would be important to evaluate to what extent are current neural networks' representations capturing the existing features. Here we propose a way to evaluate the representation quality of neural networks using a novel type of zero-shot test, entitled Raw Zero-Shot. The main idea lies in the fact that some features are present on unknown classes and that unknown classes can be defined as a combination of previous learned features. To evaluate the soft-labels of unknown classes, two metrics are proposed. One is based on clustering validation techniques (Davies-Bouldin Index) and the other is based on soft-label distance of a given correct soft-label. Experiments show that such metrics are in accordance with the robustness to adversarial attacks and might serve as a guidance to build better models as well as be used in loss functions to create new types of neural networks. Interestingly, the results suggests that dynamic routing networks such as CapsNet have better representation while current deeper DNNs are trading off representation quality for accuracy.

## 1 INTRODUCTION

Adversarial samples are slightly perturbed inputs that can make neural networks misclassify. They are carefully crafted by searching for variations in the input that, for example, could decrease the soft-labels of the correct class. Since they were discovered some years ago (Szegedy, 2014), the number of adversarial samples have grown in both number and types. Random noise was shown to be recognized with high confidence by neural networks (Nguyen et al., 2015), universal perturbations, that can be added to almost any image to generate an adversarial sample, were shown to exist (Moosavi-Dezfooli et al., 2017), and the addition of crafted patches was shown to cause networks to misclassify (Brown et al., 2017). Only one pixel is enough to make networks misclassify (Su et al., 2017). Such attacks can also be easily transferred to real-world scenarios (Kurakin et al., 2016),(Athalye & Sutskever, 2018), which confers a big issue as well as a security risk for current deep neural networks' applications.

Albeit the existence of many defences, there is not any known learning algorithm or procedure that can defend against adversarial attacks consistently. Many works have tried to defend by hiding or modifying the gradients to make neural networks harder to attack. However, a recent paper shows that most of these defences fall into the class of obfuscated gradients which have their shortcomings (e.g., they can be easily bypassed by transferable attacks) (Athalye et al., 2018). Additionally, the use of an augmented dataset with adversarial samples (named adversarial training) is perhaps one of the most successful approaches to construct robust neural networks (Goodfellow et al., 2014),(Huang et al., 2015), (Madry et al., 2018). However, it is still vulnerable to attacks and has a strong bias in the type of adversarial samples used in training (Tramèr et al., 2018).

This shows that a deeper understanding of the issues is needed to enable more consistent defences to be created. Few works focused on understanding the reason behind such lack of robustness. In (Goodfellow et al., 2014), it is argued that Deep Neural Networks's (DNN) linearity are one of the main reasons. Recent investigations reveal that attacks are changing where the algorithm is paying attention (Vargas & Su, 2019), other experiments show that deep learning neural networks learn false

structures that are easier to learn rather than the ones expected (Thesing et al., 2019) and an accuracy and robustness trade-off for models were shown to exist (Tsipras et al., 2019).

In this paper, we propose a methodology of how to evaluate the representation of machine learning methods. Based on these metrics, we reveal a link between deep representations' quality and attack susceptibility. Specifically, we propose a test called Raw Zero-Shot and two metrics to evaluate DNN's representations.

## 1.1 RECENT ADVANCES IN ATTACKS AND DEFENSES

DNNs were shown vulnerable to many types of attacks. For example, the output high confidence results to noise images (Nguyen et al., 2015), universal perturbations in which a single perturbation can be added to almost any input to create an adversarial sample are possible (Moosavi-Dezfooli et al., 2017), the addition of image patches can also make them misclassify (Brown et al., 2017). Moreover, the vulnerability can be exploited even with a single pixel, i.e., changing a single pixel is often enough to make a DNNs misclassify (Su et al., 2017). Most of these attacks can be transformed into real-world attacks by merely printing the adversarial samples (Kurakin et al., 2016). Moreover, crafted glasses (Sharif et al., 2016) or even general 3d adversarial objects (Athalye & Sutskever, 2018) can be used as attacks.

Although many defensive systems were proposed to tackle the current problems, there is still no consistent solution available. Defensive distillation in which a smaller neural network squeezes the content learned by the original DNN was proposed (Papernot et al., 2016). However, it was shown not to be robust enough (Carlini & Wagner, 2017). Adversarial training was also proposed as a defence, in which adversarial samples are used to augment the training dataset (Goodfellow et al., 2014),(Huang et al., 2015), (Madry et al., 2018). With adversarial training, DNNs increase slightly in robustness but not without a bias towards the adversarial samples used and while still being vulnerable to attacks in general (Tramèr et al., 2018). There are many recent variations of defenses in which the objective is to hide the gradients (obfuscated gradients) (Ma et al., 2018), (Guo et al., 2018) (Song et al., 2018). However, they can be bypassed by various types of attacks (such as attacks not using gradients, transfer of adversarial samples, etc.) (Athalye et al., 2018),(Uesato et al., 2018).

There are a couple of works which are trying to understand the reason behind such lack of robustness. To citep some, in (Goodfellow et al., 2014), it is argued that the main reason may lie in DNNs' lack of non-linearity. Another work argues that the perturbations cause a change in the saliency of images which makes the model switch the attention to another part of it (Vargas & Su, 2019). False structures that are easier to learn were also shown related to the problem (Thesing et al., 2019). Moreover, in (Tsipras et al., 2019), the accuracy and robustness trade-off was shown to exist.

## 1.2 ZERO-SHOT LEARNING

Zero-Shot learning is a method used to estimate unknown classes which do not appear in the training data. The motivation of Zero-Shot learning is to transfer knowledge from training classes to unknown classes. Existing methods approach the problem by estimating unknown classes from an attribute vector defined manually. Attribute vectors are annotated to both known and unknown classes, and for each class, whether an attribute, such as "colour" and "shape", belongs to the class or not is represented by 1 or 0. For example, in (Lampert et al., 2009) the authors proposed *Direct Attribute Prediction (DAP)* model, which learns each parameter for estimating the attributes from the target data. It estimates an unknown class of the source data which is estimated from the target data by using these parameters. Based on this research, other zero-shot learning methods have been proposed which uses an embedded representation generated using a natural language processing algorithm instead of a manually created attribute vector (Zhang & Saligrama, 2016; Fu et al., 2015; Norouzi et al., 2013; Akata et al., 2015; Bucher et al., 2016).

In (Zhang & Saligrama, 2015), a different approach to estimate unknown classes is proposed. This method constructs the histogram of known classes distribution for an unknown class. In this approach, it is assumed that the unknown classes are the same if these histograms generated in the target domain and the source domain are similar. This perspective is similar to our approach because our method approach to represent an unknown class as the distribution of known classes. However, our objective

is not estimating the unknown class, and we do not use the source domain. Our objective here is to analyze DNNs' representation by using this distribution.

## 2 ON THE LINKS BETWEEN ROBUSTNESS EVALUATION AND REPRESENTATION

In the canonical classification setting, the goal of a classifier is to achieve low expected loss:

$$\mathbb{E}_{(x,y)\sim D}[\mathcal{L}(x,y;\theta)] \tag{1}$$

Robustness against adversarial attacks is a slightly different setting. To achieve a high robustness in this setting, a classifier should have a lower adversarial loss in noise $\delta \in \Delta$ [1]:

$$\mathbb{E}_{(x,y)\sim D}[\mathcal{L}(x+\delta,y;\theta)]. \tag{2}$$

Considering Mean Squared Error (MSE), we have:

$$\mathbb{E}_{(x,y)\sim D}[\mathcal{L}(x+\delta,y;\theta)] = \mathbb{E}_{(x,y)\sim D}[(f(x+\delta)-h(x+\delta))^2] + \mathbb{E}_{(x,y)\sim D}[(h(x+\delta)-\hat{y}(x+\delta))^2],$$

where $h(x) = E[h_D(x)]$ is the expected behavior of the prediction when averaged over many datasets, $f(x)$ is the ground-truth and $\hat{y}(x) = h_D(x)$ is the output after learned on a given dataset $D$. For robustness to increase, adversarial training requires that datasets should have many noisy samples, i.e., $x+\delta \in D$. However, the more noise is added to images the more $D$ becomes close to all possible images $\mathbb{R}^{M*N}$:

$$\lim_{\Delta \to \infty} D = \mathbb{R}^{M*N} \tag{3}$$

However, $f(x)$ is undefined [2] for $D \in \mathbb{R}^{M*N}$ in which $y \notin C$ for the set of known classes $C$. Even a small amount of noise may be enough to cause $y \notin C$ and thus $f(x)$ undefined. Would it be possible to evaluate the robustness and/or the quality of a model without a well defined $y$?

To answer this question we take into account an ideal representation $z$ and the representation learned by the model $\hat{z}$:

$$\mathbb{E}[(f(x+\delta;z)-h(x+\delta;\hat{z}))^2] + \mathbb{E}[(h(x+\delta;\hat{z})-\hat{y}(x+\delta;\hat{z}))^2]$$

Interestingly, although $y$ is undefined, $z$ represents the features learned and is well defined for any input. Moreover, by considering learned classes to be clustered in $z$ space, unsupervised learning evaluation can be used to evaluate $z$ even without a well defined $y$. We use here a famous clustering analysis index to evaluate clusters in $z$ by their intracluster distance. In $z$, it is also possible to evaluate the representation of known and unknown classes which should share some features. Moreover, we hypothesize here that unknown classes should evaluate $z$ with less bias because a direct map from input to output is inexistent. Any projection of the input in any of the feature maps or the output layer could be used as $z$. To take the entire projection into account, we use here $z$ as the final projection of the input to the classes, i.e. $z$ is the soft label array **e**.

## 3 RAW ZERO-SHOT

In this paper, we propose to evaluate the representation learned by conducting experiments over the soft-labels of the image in unknown classes. This is based on the hypothesis that if a model is

---

[1]Here we use the error in noise to be the adversarial loss instead of the worst-case error, for a discussion of the relationship between error in noise and adversarial samples please refer to (Gilmer et al., 2019)

[2]Alternatively, $f(x)$ could be defined for any noise if an additional unknown class is defined, such as with an OpenMax layer (Bendale & Boult, 2016).

capable of learning useful features, an unknown class would also trigger some of these features inside the model. We call this type of test over unknown classes and without any other information, Raw Zero-Shot (Figure 1).

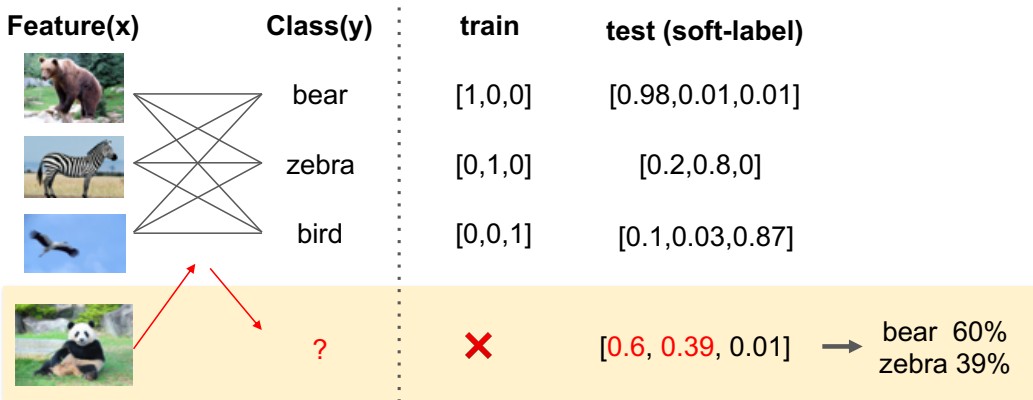

Figure 1: Raw Zero-Shot Illustration.

The Raw Zero-Shot is a supervised learning test in which only $n - 1$ of the $n$ classes are shown to the classifier during training. The classifier also has only $n - 1$ possible outputs. During testing, only unknown classes are presented to the classifier. The soft-labels outputted for the given unknown class is recorded, and the process is repeated for all possible $n$ classes, removing a different class each time.

To evaluate the representation quality, metrics computed over the soft-labels are used. These metrics are based on a different hypothesis of what defines a feature or a class. In the same way that there are different types of robustness, there are also different types of representation quality. Therefore, metrics are somewhat complementary, each highlighting a different aspect of the whole. The following subsections define two of them.

### 3.1 DAVIES-BOULDIN METRIC - CLUSTERING HYPOTHESIS

Soft labels of a classifier compose a space in which a given image would be classified as a weighted vector concerning the previous classes learned. Considering that a cluster in this space would constitute a class, we can use clustering validation techniques to evaluate the representation (Figure 2).

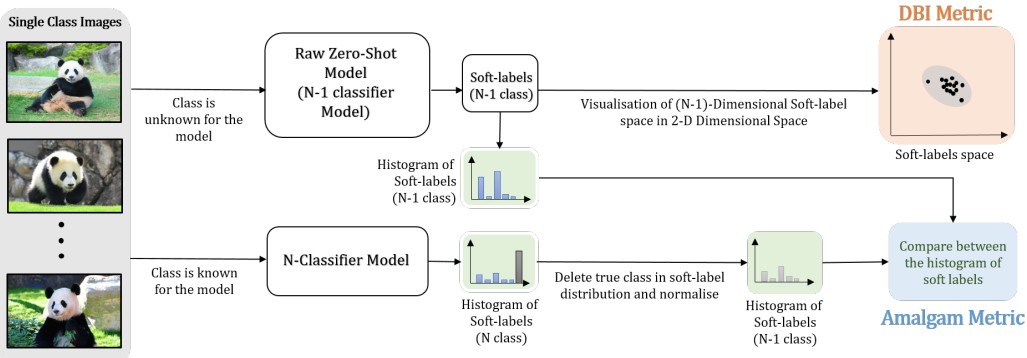

Figure 2: Illustration of both DBI and Amalgam metrics.

Here we choose for simplicity one of the most used metric in internal cluster validation, Davies-Bouldin Index (DBI). DBI is defined as follows:

$$DBI = \left( \frac{1}{n_e} \sum_{j=1}^{n_e} |\mathbf{e}_j - \mathbf{cn}|^2 \right)^{1/2}, \tag{4}$$

in which $\mathbf{cn}$ is the centroid of the cluster, $\mathbf{e}$ is one soft-label and $n_e$ is the number of samples.

## 3.2 Amalgam Metric - Amalgam Hypothesis

If DNNs can learn the features present in the classes, it would be reasonable to consider that the sof-labels also describe a given image as a combination of the previously learned classes. This is also true when an image contains an unknown class. Similar to a vector space in linear algebra, the soft-labels can be combined to describe unknown objects in this space. This is analogous to how children describe previously unseen objects as a combination of previously seen objects. Differently from the previous metric, here we are interested in the exact values of the soft-labels. However, what would constitute the correct soft-labels for a given unknown class needs to be determined.

To calculate the correct soft-label of a given unknown class (amalgam proportion) automatically, we use here the assumption that accurate classifiers should output a good approximation of the amalgam proportion already. Therefore, if a classifier is trained in the $n$ classes, the soft-labels of the remaining $n-1$ classes is the amalgam proportion (Figure 2 illustrates the concept). Consequently, the Amalgam Metric (AM) is defined as:

$$\begin{aligned} \mathbf{h'}_i &= \sum_{j=1}^{n_e} \mathbf{e'}_j, \mathbf{h}_i = \sum_{j=1}^{n_e} \mathbf{e}_j \\ AM &= \left( \frac{1}{n} \sum_{i=1}^{n} \frac{\|\mathbf{h'}_i - \mathbf{h}_i\|_1}{n-1} \right), \end{aligned} \tag{5}$$

in which, $\mathbf{e'}$ is the normalized (such that they sum to one) soft-label from the classifier trained over $n$ classes and $\mathbf{e}$ is the soft-labels from the classifier trained over $n-1$ classes.

## 4 Raw Zero-Shot Experiments

Here, we conduct Raw Zero-Shot experiments to evaluate the representation of DNNs. To obtain results over a wide range of architectures, we chose to evaluate CapsNet (a recently proposed completely different architecture based on dynamic routing and capsules) (Sabour et al., 2017), ResNet (a state-of-the-art architecture based on skip connections)(He et al., 2016), Network in Network (NIN) (an architecture which uses micro neural networks instead of linear filters) (Lin et al., 2013), All Convolutional Network (AllConv) (an architecture without max pooling and fully connected layers)(Springenberg et al., 2015) and LeNet (a simpler architecture which is also a historical mark) (LeCun et al., 1998). All the experiments are run over the CIFAR dataset by using a training dataset with all the samples of one specific class removed. This process is repeated for all classes, removing the samples of a different class each time.

To analyze the correlation between representation metrics and robustness against adversarial attacks, we conducted adversarial attacks on all the architectures tested using the most well known algorithms such as Carlini (Carlini & Wagner, 2017), Fast Gradient Method (FGM) (Goodfellow et al., 2014), Basic Iterative Method (BIM )(Kurakin et al., 2016), DeepFool (Moosavi-Dezfooli et al., 2016), Projected Gradient Descent Method (PGDM) (Madry et al., 2018) (Table 1).

For all tests, $\varepsilon$ is fixed to the corresponding value given in the table. However, different methods have different meanings for $\varepsilon$: (a) for pixel attacks, $\varepsilon$ is the maximum number of pixels to be changed, (b) for threshold attacks, $\varepsilon$ is the maximum amount of change allowed per pixel, (c) for FGM, BIM and PGDM, $\varepsilon$ is the attack step size (input variation) and (d) for DeepFool, $\varepsilon$ is the overshoot parameter.

Table 2 shows the ranking based on the attack accuracy and their required perturbation (Table 1). In general, CapsNet is shown to be the most robust, AllConv and DenseNet follow with a proper

| Model | WideResNet | DenseNet | ResNet | NetInNet | AllConv | CapsNet | LeNet |
|---|---|---|---|---|---|---|---|
| **Accuracy (in %)** | | | | | | | |
| NewtonFool | 51.00 | 50.17 | 51.80 | 53.45 | 62.33 | 75.00 | 66.24 |
| PGD | 94.90 | 94.10 | 94.34 | 91.84 | 89.24 | 79.31 | 88.78 |
| Virtual | 39.14 | 43.68 | 46.05 | 47.07 | 56.72 | 52.27 | 58.87 |
| FGM | 78.69 | 78.19 | 81.91 | 83.11 | 81.32 | 87.54 | 85.28 |
| Deep Fool | 44.47 | 48.00 | 49.32 | 51.91 | 62.76 | 68.75 | 61.91 |
| BIM | 98.18 | 97.98 | 97.65 | 97.26 | 96.95 | 96.70 | 94.12 |
| Carlini | 94.39 | 65.50 | 95.99 | 86.44 | 88.68 | 57.64 | 77.30 |
| *Average* | *71.54* | *68.23* | *73.87* | *73.01* | *76.86* | *73.89* | *76.07* |
| **L2 Score** | | | | | | | |
| NewtonFool | 2564.0167 | 2675.4007 | 2657.9091 | 2556.8375 | 2555.9087 | 2451.6315 | 2518.5794 |
| PGD | 935.1888 | 934.5662 | 934.9485 | 942.2018 | 946.7355 | 962.7172 | 952.4294 |
| Virtual | 2656.4656 | 2673.4656 | 2629.6419 | 2631.2242 | 2620.5943 | 2595.9619 | 2579.7693 |
| FGM | 2497.5955 | 2534.9078 | 2504.6836 | 2515.7904 | 2523.1862 | 2506.7086 | 2509.134 |
| Deep Fool | 2592.1814 | 2627.0271 | 2602.9897 | 2583.3581 | 2582.5377 | 2490.3162 | 2568.3051 |
| BIM | 3328.8821 | 3296.0967 | 3295.5864 | 3324.4652 | 3404.1802 | 3312.3621 | 3566.806 |
| Carlini | 2422.023 | 2549.4311 | 2430.7893 | 2412.361 | 2411.7259 | 2345.2674 | 2377.3322 |
| *Average* | *2428.0504* | *2470.1279* | *2436.6498* | *2423.7483* | *2434.9812* | *2380.7093* | *2438.9079* |

Table 1: Attack Accuracy (percentage of successful attacks on correct classified samples) and Mean L2 score (L2 difference between original sample and adversarial one) for the CIFAR-10 test dataset for the different architectures.

| Model | Rank (Acc) | Rank ($L_2$) | Diff |
|---|---|---|---|
| WideResNet | 3 | 7 | *4* |
| DenseNet | 2 | 4 | *2* |
| ResNet | 7 | 6 | *1* |
| NIN | 4 | 5 | *1* |
| AllConv | 5 | 2 | *3* |
| CapsNet | 1 | 1 | *0* |
| LeNet | 6 | 3 | *3* |

Table 2: Overall ranking for both accuracy (Acc) and $L_2$ attacks. The rankings are obtained by ordering the average accuracy and $L_2$ for all attacks.

placement while the remaining networks vary depending on the perspective used to analyze (i.e., higher accuracy or lower $L_2$). These robustness rankings will be used in the next sections to verify the relationship of robustness against adversarial samples and metrics to evaluate the representation quality.

## 4.1 EXPERIMENTS ON DBI METRIC

Table 3 shows the results with the DBI metric (the smallest the better) and the respective ranking of each neural network. According to this metric, CapsNet possesses the best representation of all networks tested. LeNet is considered the second-best neural network regarding representation, followed by AllConv, NIN and then deeper neural networks. DBI metric matches exceptionally well with both accuracy and $L_2$ ranking based on robustness against adversarial samples. Most differences in Hamming distance lies in the exact places in which both accuracy and $L_2$ rankings differ. Notice that DBI metric does not use anything related to attacks and still arrives at similar rankings. To further demonstrate the correlation between DBI and adversarial attacks, a Pearson correlation of the DBI for each network is shown in Table 4. This table suggests that DBI and adversarial attacks have a statistical significant correlation.

The fact that LeNet and other relatively more uncomplicated networks achieve a high representation quality which is at odds with accuracy may seem extremely unlikely. However, as discussed in (Tsipras et al., 2019), accurate models can trade-off robustness for accuracy. DBI suggests that this trade-off happens because the representation quality has worsened. Interestingly, LeNet and other

| Model | DBI | Ranking | Diff (Acc) | Diff ($L_2$) | AM | Ranking | Diff (Acc) | Diff ($L_2$) |
|---|---|---|---|---|---|---|---|---|
| WideResNet | 0.58±0.14 | 7 | 4 | 0 | 249.48±135.60 | 7 | 4 | 0 |
| DenseNet | 0.60±0.14 | 6 | 4 | 2 | 296.76±100.63 | 5 | 3 | 1 |
| ResNet | 0.63±0.13 | 5 | 2 | 1 | 281.29±107.82 | 6 | 1 | 0 |
| NIN | 0.62±0.09 | 3 | 1 | 2 | 203.15±93.08 | 4 | 0 | 1 |
| AllConv | 0.64±0.10 | 4 | 1 | 2 | 101.12±54.98 | 1 | 4 | 1 |
| CapsNet | 0.23±0.01 | 1 | 0 | 0 | 124.48±62.23 | 2 | 1 | 1 |
| LeNet | 0.51±0.02 | 2 | 4 | 1 | 144.10±75.18 | 3 | 3 | 0 |

Table 3: Mean DBI, the rank regarding this representation metric and the Hamming distance to the robustness rankings against adversarial samples for each neural network. AM value for each of the different architectures and their respective ranking. The Hamming distance of the DBI's and AM's rankings for both the accuracy and $L_2$ robustness ranking against adversarial samples are also shown.

| Model | Newton Fool | PGD | Virtual | FGM | Deep Fool | BIM | Carlini |
|---|---|---|---|---|---|---|---|
| | | | Pearson Correlation between DBI and Accuracy (p-value) | | | | |
| WideResNet | -0.48 (0.16) | 0.01 (0.98) | -0.43 (0.22) | -0.35 (0.32) | -0.42 (0.23) | -0.00 (0.99) | -0.21 (0.57) |
| DenseNet | -0.51 (0.13) | -0.19 (0.60) | -0.51 (0.13) | -0.54 (0.11) | -0.51 (0.13) | -0.28 (0.43) | -0.51 (0.13) |
| ResNet | -0.29 (0.41) | -0.30 (0.40) | -0.29 (0.42) | -0.24 (0.50) | -0.28 (0.44) | -0.26 (0.46) | -0.18 (0.62) |
| NetInNet | -0.49 (0.15) | -0.41 (0.24) | -0.51 (0.14) | -0.38 (0.27) | -0.54 (0.11) | -0.68 (0.03) | -0.50 (0.14) |
| AllConv | -0.61 (0.06) | -0.27 (0.45) | -0.64 (0.05) | -0.45 (0.20) | -0.68 (0.03) | -0.60 (0.07) | -0.70 (0.02) |
| CapsNet | -0.71 (0.02) | 0.12 (0.75) | -0.80 (0.01) | -0.45 (0.20) | -0.70 (0.02) | -0.12 (0.75) | -0.70 (0.02) |
| LeNet | -0.60 (0.06) | -0.00 (0.99) | -0.54 (0.10) | -0.28 (0.43) | -0.58 (0.08) | -0.13 (0.72) | -0.61 (0.06) |
| | | | Pearson Correlation between DBI and L2 Score (p-value) | | | | |
| WideResNet | -0.58 (0.08) | -0.61 (0.06) | -0.62 (0.06) | -0.64 (0.05) | -0.62 (0.06) | -0.68 (0.03) | -0.69 (0.03) |
| DenseNet | -0.51 (0.13) | -0.68 (0.03) | -0.58 (0.08) | -0.71 (0.02) | -0.61 (0.06) | -0.76 (0.01) | -0.74 (0.02) |
| ResNet | -0.55 (0.10) | -0.60 (0.06) | -0.56 (0.09) | -0.55 (0.10) | -0.55 (0.10) | -0.65 (0.04) | -0.66 (0.04) |
| NetInNet | -0.64 (0.05) | -0.71 (0.02) | -0.59 (0.07) | -0.66 (0.04) | -0.61 (0.06) | -0.74 (0.01) | -0.76 (0.01) |
| AllConv | -0.70 (0.02) | -0.65 (0.04) | -0.67 (0.03) | -0.74 (0.01) | -0.71 (0.02) | -0.75 (0.01) | -0.75 (0.01) |
| CapsNet | -0.80 (0.01) | -0.78 (0.01) | -0.81 (0.00) | -0.81 (0.00) | -0.76 (0.01) | -0.74 (0.01) | -0.73 (0.02) |
| LeNet | -0.57 (0.08) | -0.59 (0.07) | -0.57 (0.09) | -0.60 (0.07) | -0.56 (0.09) | -0.56 (0.09) | -0.60 (0.07) |

Table 4: Pearson correlation between DBI and both L2 score as well as accuracy of attacks

simple networks are also easier to attack (low-rank accuracy) but need more perturbation to achieve the same accuracy (high rank $L_2$). Therefore, LeNet and other simple networks might be easier to attack because the search space is less complicated (less obfuscation (Athalye et al., 2018)). However, this does not mean they are less robust. Alternatively, as DBI suggests, LeNet and other simple networks might have achieved relatively good representations but without high accuracy.

To enable visualization of this metric, we plotted in Figure 3 a projection into two dimensions of all the points in the decision space of unknown classes. All this is done while preserving the high-dimensional distance between the points. Here we use the Isomap (Tenenbaum et al., 2000) to achieve this effect. It can be easily observed that CapsNet's results for unknown classes are more clustered and thus form a better-defined cluster than other architectures.

## 4.2 Experiments on Amalgam Metric

In this section, the AM for all the networks is evaluated, which is based on the similarity of soft-labels for networks that were trained in all classes. The results shown in Table 3 reveal almost the same representation ranking as the robustness rankings related to $L_2$. Interestingly, although both DBI and AM differ widely in concept and calculation procedure, the rankings are both similar and close to the $L_2$. This further suggests that both metrics agree on what would be a good representation quality and can be used to evaluate representation in newer methods. A visualization as well as the Pearson correlation of the metric is included in the supplementary works.

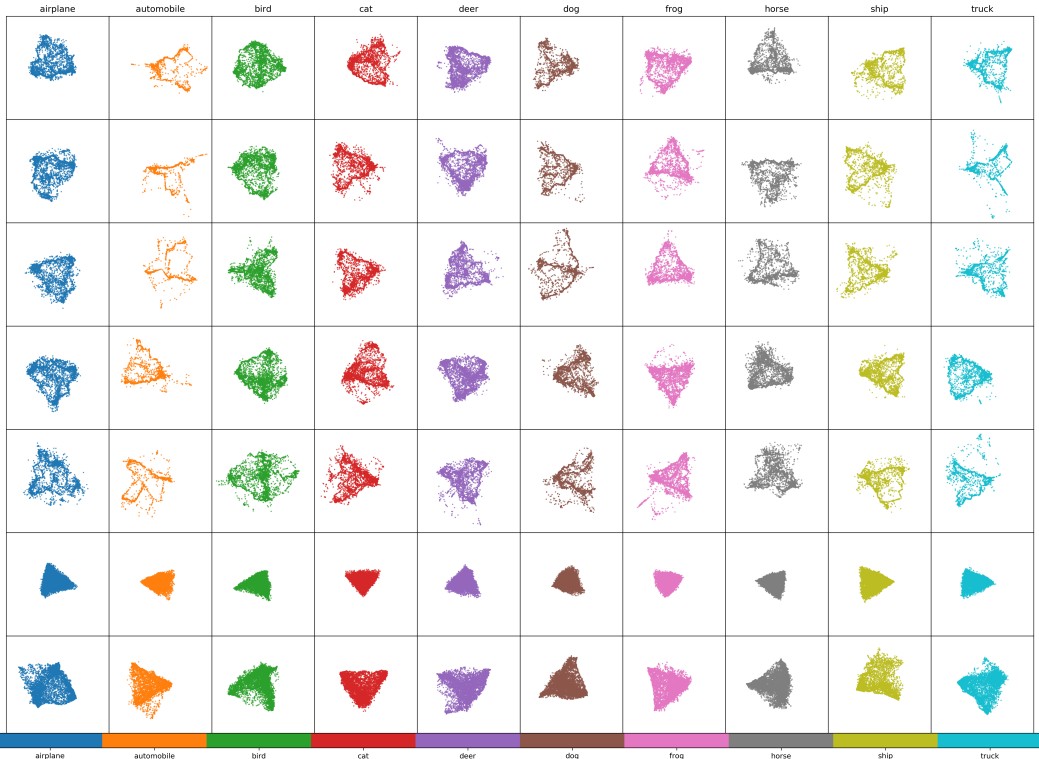

Figure 3: Visualization of the results in Table 3 using a topology preserving two-dimensional projection with Isomap. Each row shows the Isomap of one architecture. From top to bottom: WideResNet, DenseNet, ResNet, NIN, AllConv, CapsNet, LeNet.

### 4.3 PARTS OF A WHOLE

In (Agarwal et al., 2019), adding a loss to force features to be close to the feature centroid was shown to be beneficial against adversarial attacks. This is consistent with the proposed DBI metric, demonstrating that both soft-label space as well as other feature spaces benefit from projections to nearby positions. At the same time, we further support the existence of a trade-off between accuracy and robustness for deeper DNNs first point out in (Tsipras et al., 2019). Other types of networks, such as CapsNet seem to avoid it to some extent. Therefore, the trade-off is shown to vary with the architecture and computing dynamics of a model. Lastly, we hypothesize that a representation bias may hold in which features learned are only invariant to the dataset and not unseen classes.

## 5 CONCLUSIONS

Here we proposed the Raw Zero-Shot method to evaluate the representation of classifiers. In order to score the soft-labels, two metrics were formally defined based on a different hypothesis of representation quality. Results suggest that the evaluation of the representation of both metrics (DBI and AM) are linked with the robustness of neural networks. In other words, quickly attacked neural networks have a lower representation score. Interestingly, LeNet scores well in both metrics, albeit being the least accurate. LeNet is followed up by AllConv and NIN, which are less complicated/profound than other models which suggest that deeper architectures might be trading-off representation quality for accuracy. These results shown here further support the claim that there is a trade-off between accuracy and robustness in current deep learning (Tsipras et al., 2019).

Thus, the proposed Raw Zero-Shot was able to evaluate the representation quality of state-of-the-art DNNs and show their shortcomings concerning adversarial attacks, explaining many of the current problems. It also opens up new possibilities for both the evaluation (i.e. as a quality assessment) and the development (e.g., as a loss function) of neural networks.

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

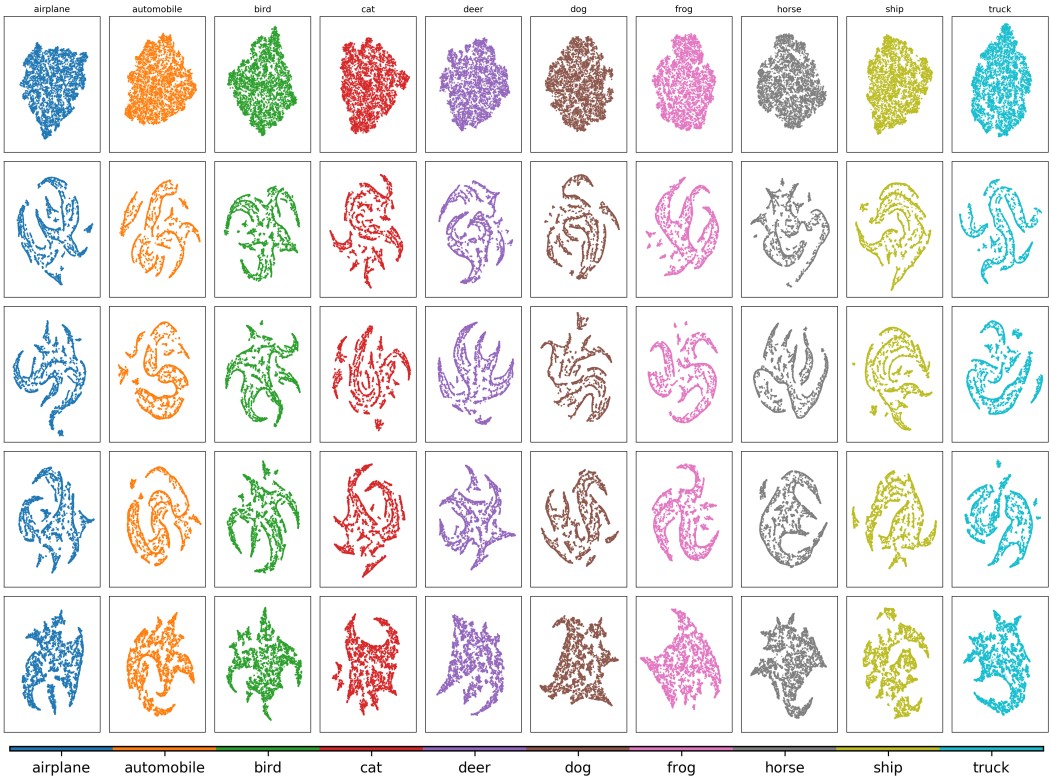

Figure 4: Visualization of the DBI Metric with t-Distributed Stochastic Neighbour Embedding (t-SNE) which focuses on the neighbour distances. Each row shows the t-SNE projections in two dimensional space for one architecture. From top to bottom: CapsNet, AllConv, ResNet, NIN and LeNet.

## SUPPLEMENTARY WORK

## A  THE POSSIBILITY OF A REPRESENTATION BIAS

The experiments show the possibility of representations to not work well for unknown classes. Many of these classes, however, share similar representations such as the dog and cat, truck and car. Thus, here we formulate a possible interpretation for the results.

The objective of a supervised learning algorithm is perhaps to map the input to output in such a way that the decision boundary reflects the real decision boundary. To achieve this, it is known that when dealing with complex problems, learning algorithms need first to learn a set of invariant features that are present throughout classes such that their recognition becomes robust against variations in the dataset.

Human beings, however, learn a set of invariant features that is not only able to solve current tasks or recognize current classes. We learn a set of features that can describe most if not all unseen classes and unknown tasks. Thus, we define representation bias as the bias towards invariant features that describe current seen classes or tasks but fail to describe unknown classes and tasks.

## B  EXTENDED ANALYSIS OF DBI METRIC

Figure 4 shows a visualization of DBI's results with t-Distributed Stochastic Neighbour Embedding (t-SNE) (Maaten & Hinton, 2008). DBI results are visualized using a projection into two dimensions while focusing on neighbour distances. The idea for having this visualisation is to investigate whether

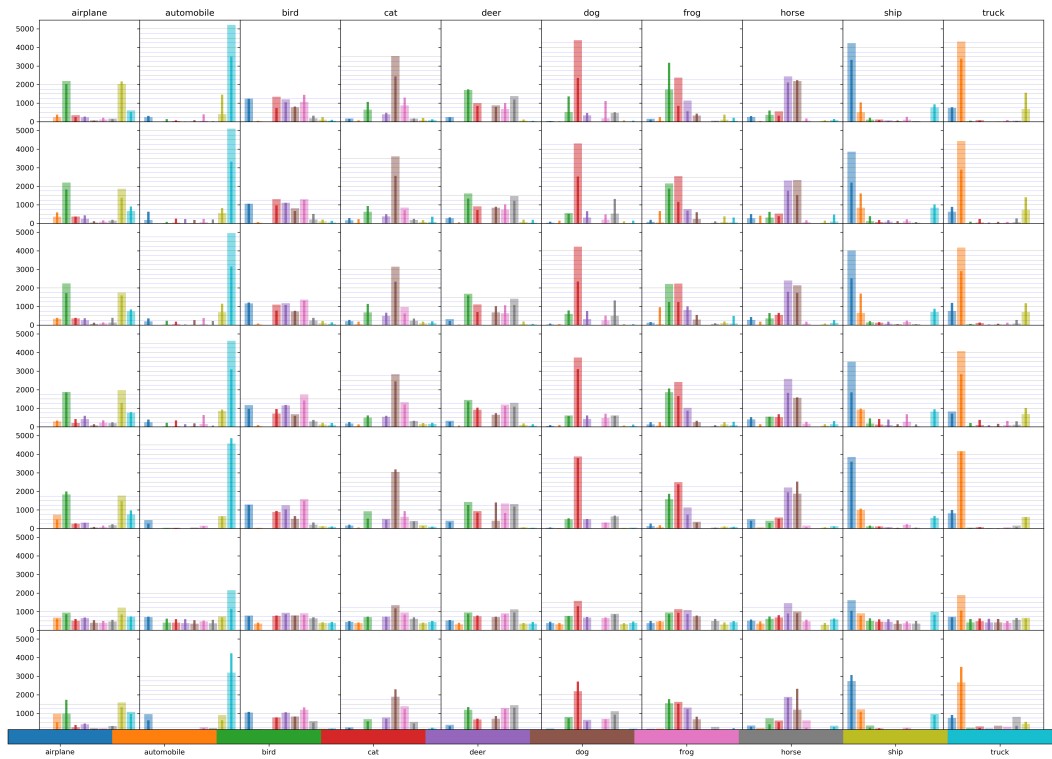

Figure 5: Histograms from which the AM is calculated. Each row shows the histograms of one architecture. Dark shaded thinner and light shaded wider bins are respectively the soft-labels from the ground-truth from the classifier trained on all classes and the soft-labels of the classifier trained on $n - 1$ classes. From top to bottom: WideResNet, DenseNet, ResNet, NIN, AllConv, CapsNet, LeNet.

the cluster can be split into one or more different classes. It is to be noted that IsoMap and t-SNE are two different visualisations for the same feature space.

The figure 4 shows that CapsNet forms pretty dense projections. Even if we form clusters to have different classes, the gaps between the classes will be too low relatively to other architectures. It also shows that in the high-dimensional space all the logits are moderately close to each other. While for the other architectures, there exists some points which can form their own cluster and be termed as different label. Hence for these architectures it can have one or more different classes for the same label which is contradicting to our hypothesis that there should exist only a single class for a single label. This inference can also be linked to adversarial attacks and supports the argument that CapsNet is relatively more robust than the other architectures.

## C  EXTENDED ANALYSIS OF AMALGAM METRIC

To enable a visualisation of the metric, the computed histograms ($\mathbf{h'}_i$ and $\mathbf{h}_i$ from Equation 5) are plotted in Figure 5. It is interesting to note that in Figure 5, the histograms from CapsNet are different from the other ones, by the entirely different architecture employed by CapsNet. This reveals that this metric can capture such representation differences. Table 5 reveals that Amalgam metric is less reliable than DBI. Its p-values are also high enough to render its relationship less obvious. Recall, however, that the ranking based on Amalgam is very similar to the raking obtained with DBI metric.

$$D = |\,\mathbf{h'}_i - \mathbf{h}_i\,|, \quad where$$
$$\mathbf{h'}_i = \sum_{j=1}^{n_e} \mathbf{e'}_j \quad and \quad \mathbf{h}_i = \sum_{j=1}^{n_e} \mathbf{e}_j \tag{6}$$

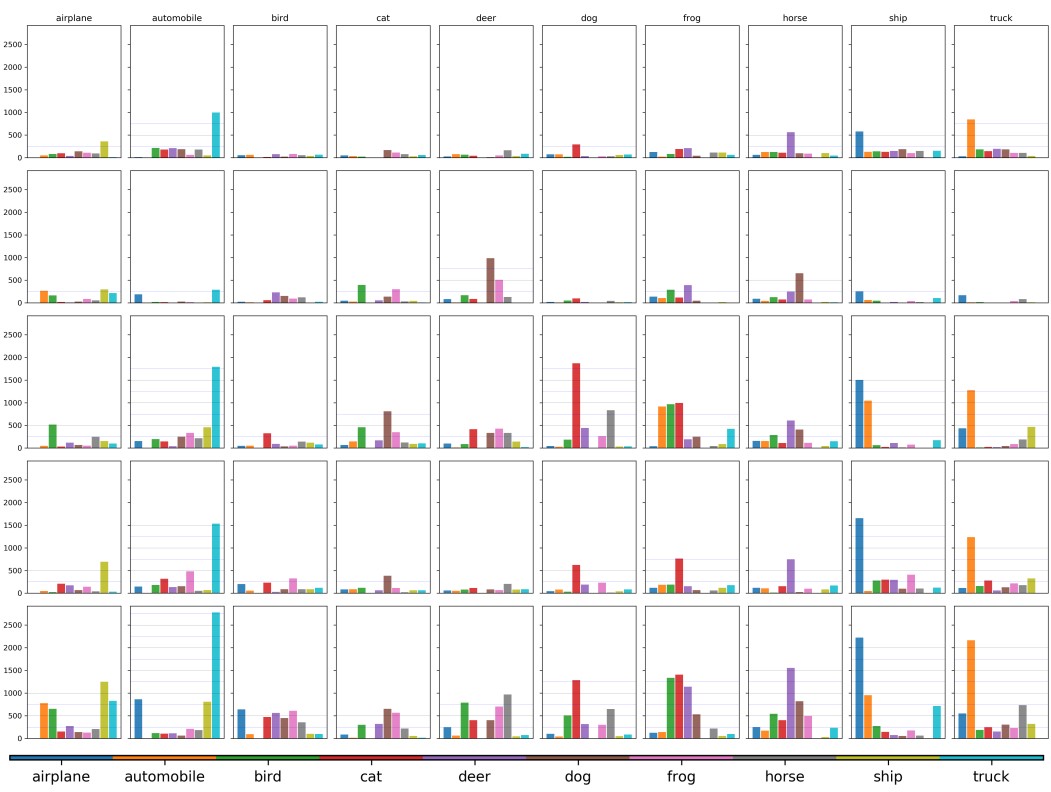

Figure 6: Histograms of $D$ for each soft label. Each row shows the difference in histograms for one architecture. From top to bottom: CapsNet, AllConv, ResNet, NIN and LeNet.

| Model | Newton Fool | PGD | Virtual | FGM | Deep Fool | BIM | Carlini |
|---|---|---|---|---|---|---|---|
| | Pearson Correlation between Amalgam and Accuracy (p-value) | | | | | | |
| WideResNet | 0.29 (0.41) | -0.05 (0.88) | 0.36 (0.31) | -0.11 (0.77) | 0.32 (0.37) | -0.22 (0.53) | 0.44 (0.20) |
| DenseNet | 0.49 (0.15) | 0.22 (0.55) | 0.49 (0.16) | 0.52 (0.13) | 0.49 (0.15) | -0.26 (0.46) | 0.51 (0.14) |
| ResNet | 0.09 (0.80) | 0.24 (0.51) | 0.09 (0.80) | 0.10 (0.78) | 0.09 (0.81) | 0.24 (0.51) | 0.04 (0.91) |
| NetInNet | -0.02 (0.95) | 0.70 (0.02) | -0.03 (0.93) | 0.12 (0.75) | 0.02 (0.96) | 0.71 (0.02) | 0.12 (0.75) |
| AllConv | -0.04 (0.91) | -0.32 (0.37) | -0.04 (0.91) | -0.10 (0.78) | -0.03 (0.93) | -0.27 (0.45) | 0.04 (0.91) |
| CapsNet | 0.66 (0.04) | -0.11 (0.76) | 0.68 (0.03) | 0.45 (0.20) | 0.73 (0.02) | 0.15 (0.69) | 0.71 (0.02) |
| LeNet | 0.61 (0.06) | 0.73 (0.02) | 0.54 (0.11) | 0.81 (0.00) | 0.57 (0.08) | 0.72 (0.02) | 0.61 (0.06) |
| | Pearson Correlation between Amalgam and L2 Score (p-value) | | | | | | |
| WideResNet | 0.66 (0.04) | 0.59 (0.07) | 0.59 (0.07) | 0.66 (0.04) | 0.64 (0.05) | 0.71 (0.02) | 0.69 (0.03) |
| DenseNet | 0.59 (0.07) | 0.70 (0.02) | 0.59 (0.07) | 0.72 (0.02) | 0.62 (0.05) | -0.75 (0.01) | -0.71 (0.02) |
| ResNet | 0.52 (0.12) | 0.43 (0.22) | 0.46 (0.18) | 0.39 (0.26) | 0.44 (0.15) | 0.49 (0.15) | 0.50 (0.14) |
| NetInNet | 0.35 (0.32) | 0.35 (0.32) | 0.39 (0.26) | 0.29 (0.42) | 0.37 (0.29) | 0.33 (0.35) | 0.39 (0.27) |
| AllConv | -0.08 (0.82) | -0.12 (0.74) | -0.10 (0.79) | -0.15 (0.69) | -0.12 (0.73) | -0.16 (0.66) | -0.15 (0.69) |
| CapsNet | 0.50 (0.14) | 0.80 (0.01) | 0.41 (0.24) | 0.81 (0.00) | 0.44 (0.20) | 0.67 (0.03) | 0.50 (0.14) |
| LeNet | 0.31 (0.38) | 0.47 (0.17) | 0.33 (0.35) | 0.41 (0.24) | 0.31 (0.38) | 0.45 (0.20) | 0.46 (0.18) |

Table 5: Pearson correlation between Amalgam Metric and both L2 score as well as accuracy of attacks

The Amalgam Metric showed that both CapsNet and AllConv have the best scores which is in accordance with their top robustness score. Figure 6 shows a visualization of equation 6 which is part of the main equation of Amalgam Metric. On analysing further the phenomena about absolute difference between **h'** and **h** in the Amalgam Metric or D in equation 6. It can be noted from the figure that for most labels of CapsNet and AllConv the difference is relatively low than the other architectures. This contributes to have CapsNet and AllConv to have best scores. Further investigations can be carried out to analyse the effect of a label in adversarial attack based on this figure. This can also provide insight on the labels which are robust to adversarial attacks. A further study can also be carried out to analyse the characteristics of the neural network's representation which makes a label more robust than other labels.

