# OpenReview forum: "Representation Quality Explain Adversarial Attacks"
_ICLR.cc/2020/Conference — Reject_

### Official Review · AnonReviewer1 · 2019-10-23
**Official Blind Review #1**

**Rating:** 1

**Review:**

=== A. Summary ===

This paper proposes to evaluate the representation (here, the output of 9 probabilities for 9 classes at the last softmax output layer) of an image taken from an unseen category by two metrics:
- The first metric (DBI) essentially measures how compact the learned manifold is (i.e. how nearby are the softmax probability vectors of images from an unseen category).
- The second metric basically measures how close a 9-D probability output vector ("soft labels" in the paper) of a classifier trained on the 9 classes is to a "ground-truth" which is the same 9-D vector but obtained when the classifier is trained on the full 10 CIFAR-10 classes.

The authors computed these "representation measures" for a set of 7 networks (6 convnets + CapsuleNet) trained on CIFAR-10.
Then, a set of white-box adversarial attacks were performed on these 7 networks and the results (both how accurate the classifier is, and the amount of L2 perturbations required to change a label) were used as a "robustness measures".
The authors then claim that the ranking of the 7 classifiers based on the representation measures match the rankings derived from the robustness measures.

The paper attempts to find the correlation between the representation quality of a classifier with its adversarial robustness. This general direction is important and worth pursuing!
However, the claimed correlation is very weakly supported by the evidence in the paper.

=== B. Decision ===

Reject.

The aim of connecting the representation quality with adversarial robustness is interesting!
However, this paper has several major issues:

0. The paper uses the wrong citation format of ICLR (changing from LastName et al. 2019 --> (12) ). The paper maybe 8.5 to 9 pages long if using the original format. If this format violation was accepted, it would be unfair to other submissions and be a bad example this citation violation would be OK.
1. The paper has many claims (e.g. "DBI metric matches extremely well") that were not supported by evidence or clarified.
2. The paper claims to study classifiers' "representation quality". However, all the experiments were conducted on only the 10-class CIFAR-10 dataset and the "representations" are taken at the softmax layer as opposed to some mid-CNN feature layer (e.g. the popular fc7 in AlexNet trained on the 1000-class ImageNet).
3. The key Definition in Sec. 2 that the paper hinges on has 0 references and is a debatable definition.

=== C. Suggestions for Improvement ===

- I'd advise the authors to perform careful literature review to identify where the gap to fill is before conducting the research. There is a ton of work that has been done in the intersection of zero-shot, adversarial examples, deep features, and classifier caliberation.

For example, [1] has looked at how out-of-distribution samples can be represented by the deep features of a classifier:
[1] Bendale, A., & Boult, T. E. (2016). Towards open set deep networks. In Proceedings of the IEEE conference on computer vision and pattern recognition (pp. 1563-1572).
- This work [2] has shown adding a "center loss" to force the deep features to be closer to the feature centroid (i.e. related to your DBI measure) helps improve the adversarial robustness. Indeed, the visualization in Fig. 3 is very interesting and is consistent with the result in [2].
[2] Agarwal, C., Nguyen, A., & Schonfeld, D. (2019, September). Improving Robustness to Adversarial Examples by Encouraging Discriminative Features. In 2019 IEEE International Conference on Image Processing (ICIP) (pp. 3801-3505). IEEE.

- The word "representation quality" here is confusing when it refers to the soft labels, which is more relevant to *caliberation* than quality of deep features as the title implies. And the paper misses this body of work in the related work. See here: https://nicholas.carlini.com/writing/2019/all-adversarial-example-papers.html
- Try to avoid "extremely" or "clearly" in claims (and in general scientific writing) because they tend to be overclaims and it is hard for readers to interpret those intensifiers.
- I don't see how the rankings by DBI or AM are similar to the original rankings by adversarial measures at all (except for CapsNet). At least, reporting Pearson correlation (or any statistical similarity measure) would be more convincing.
- Section 2 is highly debatable and has no references for the "Definition".
- I'd perform this work on ImageNet, at least to follow the intuition in Sec. 2. With only 10 CIFAR-10 classes, it is unrealistic how to semantically represent a "frog" using the other 9 classes (truck, airplane, etc).


**Experience Assessment:**

I have published in this field for several years.

**Review Assessment: Checking Correctness Of Derivations And Theory:**

I assessed the sensibility of the derivations and theory.

**Review Assessment: Checking Correctness Of Experiments:**

I assessed the sensibility of the experiments.

**Review Assessment: Thoroughness In Paper Reading:**

I read the paper thoroughly.

---

> ### Author Response · Authors · 2019-11-15
> **Answer to Reviewer #1**
>
>
> Thank you for your insights and detailed comments which not only improved strongly the paper but also further motivated it. It is very motivating to hear that the research direction is worthy pursuing when most of the published research have different direction from the current one.  We hope that the claims are now better supported by the additional experiments and mathematical formulation.
>
> > 0. The paper uses the wrong citation format of ICLR (changing from LastName et al. 2019 --> (12) ). The paper maybe 8.5 to 9 pages long if using the original format. If this format violation was accepted, it would be unfair to other submissions and be a bad example this citation violation would be OK.
>
> Sorry for the format discrepancy. An error introduced by a missing year in the bibtex compilation caused the issue. Corrected now.
>
> > 1. The paper has many claims (e.g. "DBI metric matches extremely well") that were not supported by evidence or clarified.
>
> All the claims not supported by experiments were removed. We also removed "extremely" or "clearly" from the claims.
>
> > 2. The paper claims to study classifiers' "representation quality". However, all the experiments were conducted on only the 10-class CIFAR-10 dataset and the "representations" are taken at the softmax layer as opposed to some mid-CNN feature layer (e.g. the popular fc7 in AlexNet trained on the 1000-class ImageNet).
>
> The representations could be taken at any portion of the neural network. However, taking the representations at earlier stages such as fc7 in AlexNet would exclude the later portions of the representation from the analysis. Therefore, we opted by the last projection of the input as the representation.
>
> > 3. The key Definition in Sec. 2 that the paper hinges on has 0 references and is a debatable definition.
>
> Section 2 changed to a mathematical formulation which explains the link between representation and robustness as well as a motivation behind going beyond losses to evaluate the robustness.
> Representation bias is a possible hypothesis and therefore we removed from most of the main paper, leaving only a brief mention in Section 4.3. The section about Representation bias was moved to the supplementary works.
>
> > - I'd advise the authors to perform careful literature review to identify where the gap to fill is before conducting the research. There is a ton of work that has been done in the intersection of zero-shot, adversarial examples, deep features, and classifier caliberation.
> For example, [1] has looked ... (pp. 3801-3505). IEEE.  (original message shortened to keep answer within the maximum allowed length)
>
> Thank you for mentioning these related papers. They were cited and discussed in the new version.
>
> > - The word "representation quality" here is confusing when it refers to the soft labels, which is more relevant to *caliberation* than quality of deep features as the title implies. And the paper misses this body of work in the related work. See here: https://nicholas.carlini.com/writing/2019/all-adversarial-example-papers.html
>
> The representation could be any layer of the neural network. However, we argue that middle layers would lose the last part of the transformation/projection that happens in the last layers which may be relevant to adversarial attacks. This explains our choice of the last layer, i.e., to capture the entire projection of the input. We added this explanation to Section 2.
>
> > - Try to avoid "extremely" or "clearly" in claims (and in general scientific writing) because they tend to be overclaims and it is hard for readers to interpret those intensifiers.
>
> All overclaims were removed accordingly.
>
>
> > - I don't see how the rankings by DBI or AM are similar to the original rankings by adversarial measures at all (except for CapsNet). At least, reporting Pearson correlation (or any statistical similarity measure) would be more convincing.
>
> We added the Pearson correlation for both DBI and Amalgam (for lack of space, Amalgam’s Pearson correlation was added in the supplementary works).
> Notice that L2 and accuracy rankings differ originally. Therefore, it is impossible to be similar at the same time to both L2 and accuracy rankings.
>
> >- Section 2 is highly debatable and has no references for the "Definition".
>
> Section 2 was moved to the supplementary works. The definition of the representation bias is mentioned albeit briefly in Section 4.3 as a hypothesis that could explain the results. All other references to representation bias were removed.
>
> > I'd perform this work on ImageNet, at least to follow the intuition in Sec. 2 ...  (message shortened)
>
> Giving the time and space (the paper has already 8 pages with some supplementary works) constraints, we were unable to perform the tests on Imagenet. However, we believe that the Pearson correlation proves the relationship while the representation bias remains as a hypothesis that needs to be confirmed.

---

### Official Review · AnonReviewer3 · 2019-10-24
**Official Blind Review #3**

**Rating:** 3

**Review:**

Summary:
This paper aims at revealing the relationship between the quality of deep representations and the attack susceptibility of deep classification models. To this end, they propose the zero-shot test to investigate the "quality" of learned representations for unknown classes. Specifically, they leverage two kinds of quality metrics on data of unknown classes. The first one is based on clustering named Davies-Bouldin Index which measures the compactness of intra-cluster. The second one is based on the difference of soft-label histogram distributions with/without unknown classes during training, which may describe the generalization for unknown classes or bias towards known classes of learned features. Finally, with these two metrics, they rank the quality of different models and compare such ranking results with the attack robustness obtained by different attack techniques on CIFAR-10 dataset.

+Strengths:
1. This paper shows satisfactory related works about recent advances in adversarial attacks and defenses.

-Weaknesses:
1. The writing of this paper contains many mistakes, especially for the issue of using singular and plural. I strongly recommend the authors to polish the paper carefully. Take the first two pages as an example, “results suggests” in Abstract, “Networks’s” in Introduction, “a DNNs”, “perturbations causes” in Sec1.1, “methods… which uses” in Sec.1.2, etc.
2. The idea of representations quality on unknown classes determining attack robustness is not reasonable. Since the adversarial attack is defined on the known classes, the reasons for that are in two aspects. The first is the training dataset. If the collected training data cannot represent/fulfil the whole continuous distribution/manifold of the categories or even biased, the models are of course easily fooled by unseen modes during test. The second is the mapping for classification is from high to low, which is naturally many to one. In a word, I think the quality of models on unknown classes is not directly related to the problem of adversarial attack.
3. This paper only conducts experiments on CIFAR-10 dataset, which is not convincing enough. It would be better to evaluate their method on more challenging benchmarks and also give more validation about their idea. E.g. on ImageNet, if enough number of categories has been seen for models, whether they would become more robust to adversarial attack. Evaluating features of more layers rather than the softmax outputs is also needed.
4. What is amalgam proportion? Please explain it in detail or give a reference paper. Otherwise the readers cannot understand the motivation of the second metric in Sec3.2. Besides, Fig.2 contains little information and few captions for readers to understand their method.
5. In Tab.1 and 2, the meanings of Attack Accuracy and Average Amount of Perturbations (typo in caption for Tab.2) are not introduced. Tab.5 shows 7 methods but its caption says “five”. Fig.4 is also confused, for AM, which histogram is with unknown classes and which one is without?
6. The title says "representation quality explain adversarial attacks". After reading this paper, I haven’t found the mechanism leading to the adversarial attacks of DNNs.

**Experience Assessment:**

I have published one or two papers in this area.

**Review Assessment: Checking Correctness Of Derivations And Theory:**

I carefully checked the derivations and theory.

**Review Assessment: Checking Correctness Of Experiments:**

I carefully checked the experiments.

**Review Assessment: Thoroughness In Paper Reading:**

I read the paper thoroughly.

---

> ### Author Response · Authors · 2019-11-15
> **Answer to Reviewer #3**
>
>
> Thank you for the comments and suggestions which enabled to further enhance the quality of the paper. We hope the new version of the paper has corrected previous pointed issues as well as improved in representation and experiments.
>
>
> > 1. The writing of this paper contains many mistakes, especially for the issue of using singular and plural. I strongly recommend the authors to polish the paper carefully. Take the first two pages as an example, “results suggests” in Abstract, “Networks’s” in Introduction, “a DNNs”, “perturbations causes” in Sec1.1, “methods… which uses” in Sec.1.2, etc.
>
> The paper was polished. The above grammar issues and other typos were removed from the final version. We apologise for any inconvenience caused by it.
>
> > 2. The idea of representations quality on unknown classes determining attack robustness is not reasonable. Since the adversarial attack is defined on the known classes, the reasons for that are in two aspects. The first is the training dataset. If the collected training data cannot represent/fulfil the whole continuous distribution/manifold of the categories or even biased, the models are of course easily fooled by unseen modes during test. The second is the mapping for classification is from high to low, which is naturally many to one. In a word, I think the quality of models on unknown classes is not directly related to the problem of adversarial attack.
>
> We do not conduct attacks on unknown classes but evaluate the quality of representation on unknown classes. By representation we mean high level features. Therefore, we want to avoid any mapping of input to output and rather investigate to what extent features are still working for different patterns with similar features.
> The presentation of the paper was not clear enough. To improve the presentation further, Section 2 was added. This section explains the mathematical motivation behind the metrics as well as the motivation behind going beyond losses to evaluate robustness on models.
>
> > 3. This paper only conducts experiments on CIFAR-10 dataset, which is not convincing enough. It would be better to evaluate their method on more challenging benchmarks and also give more validation about their idea. E.g. on ImageNet, if enough number of categories has been seen for models, whether they would become more robust to adversarial attack. Evaluating features of more layers rather than the softmax outputs is also needed.
>
> Indeed, further tests on ImageNet would be able to tell if representation improves. Having said that, the paper has already 8 pages with some supplementary works, increasing the number of tests is not possible given the space and time constraints.
>
> > 4. What is amalgam proportion? Please explain it in detail or give a reference paper. Otherwise the readers cannot understand the motivation of the second metric in Sec3.2. Besides, Fig.2 contains little information and few captions for readers to understand their method.
>
> Figure 2 was further enhanced with more details to facilitate the understanding of the Amalgam metric. Amalgam metric is the difference between the N-1 soft-labels (i.e., excluding the true class label) for a classifier that knows the class and for a classifier that does not know the class. The classifier that knows the class acts as a ground-truth.
>
> > 5. In Tab.1 and 2, the meanings of Attack Accuracy and Average Amount of Perturbations (typo in caption for Tab.2) are not introduced. Tab.5 shows 7 methods but its caption says “five”. Fig.4 is also confused, for AM, which histogram is with unknown classes and which one is without?
>
> Thank you for pointing out. Explanation added to Table 1 and 2 (now mergerd). Tab. 5 was corrected. For lack of space, Fig. 4 was fixed and moved to the appendix. Additional description was also added.
>
>
> > 6. The title says "representation quality explain adversarial attacks". After reading this paper, I haven’t found the mechanism leading to the adversarial attacks of DNNs.
>
> Section 2 now explains mathematically how robustness evaluation may need to be defined over representation in order to consider all possible types of attacks and how representation is linked to robustness evaluation.
> The experiments show that the link holds and now there is a person correlation further evidencing it.

---

### Official Review · AnonReviewer4 · 2019-10-30
**Official Blind Review #4**

**Rating:** 1

**Review:**

This paper proposes to evaluate the robustness of the neural networks by extrapolating to the unseen classes. However, the authors only include evaluation for non-robust trained models, without considering the robust trained model, such as Madry et al. [1]. The conclusion is not convincing that the authors studied the robustness using only non-robust models, because it is well known that the accuracy for attacking non-robust model can be 100% (for CIFAR-10). It is useful if the authors study whether their method can be used to measure the robustness of the robust trained models.

1. The paper only evaluate robust accuracy on models without robust training.

2. The evidence in the paper does not support their point. The paper shows that there's link between adversarial robustness and the generalization ability of neural network to extrapolate to unseen classes. But could not support the claim in the abstract, that

"The main idea lies in the fact that some features are present on unknown classes and that unknown classes can be defined as a combination of previous learned features without representation bias (a bias towards representation that maps only current set of input-outputs and their boundary)"

The extrapolation score does not prove this.

3. The attack accuracy in Table 1 is not convincing. In [1], the PGD for epsilon=0.3 is 100% for non-robust models and around 55% for robust models, the number in the Table matches non of these.

4. The writing of this paper is not clear and has gramma issues. For example, Table 1 and 2 miss information in Caption.

5. The author should cite "Adversarial Examples Are a Natural Consequence of Test Error in Noise", which also studied the generalization and adversarial robustness.

6. This evaluation measurement is not practical, it costs more computation than one evaluates a model directly using attacks. The measurement is also fragile under adversarial attacks, that one can feed in adversarial attacks to fool the score metrics, which is not convincing.

**Experience Assessment:**

I have published one or two papers in this area.

**Review Assessment: Checking Correctness Of Derivations And Theory:**

I assessed the sensibility of the derivations and theory.

**Review Assessment: Checking Correctness Of Experiments:**

I carefully checked the experiments.

**Review Assessment: Thoroughness In Paper Reading:**

I read the paper at least twice and used my best judgement in assessing the paper.

---

> ### Author Response · Authors · 2019-11-15
> **Answer to Reviewer #4**
>
> Thank you for your review and comments which helped improve the paper further.
>
> >1. The paper only evaluate robust accuracy on models without robust training.
>
> Adversarial training is specific for robustness against particular attacks and not attacks in general. For example, in Vargas and Kotyan (2019) the authors showed that L\infty based adversarial training fail to protect against L0 attacks. Therefore, adding an experiment on the adversarial training would lead to more confusion among the readers as different types of adversarial training exist and for each type a different vulnerability will be found.
>
> Vargas, D. V., & Kotyan, S. (2019). Model Agnostic Dual Quality Assessment for Adversarial Machine Learning and an Analysis of Current Neural Networks and Defenses. arXiv preprint arXiv:1906.06026.
>
> >2. The evidence in the paper does not support their point. The paper shows that there's link between adversarial robustness and the generalization ability of neural network to extrapolate to unseen classes. But could not support the claim in the abstract, that
> >"The main idea lies in the fact that some features are present on unknown classes and that unknown classes can be defined as a combination of previous learned features without representation bias (a bias towards representation that maps only current set of input-outputs and their boundary)"
> >The extrapolation score does not prove this.
>
> True, this is an interpretation of the results. It is not the only interpretation and also not proved. Therefore, we removed from the abstract and from the main text, only adding a small section talking about it on the supplementary works and a small mention of the hypothesis in Section 4.3.
>
> > 3. The attack accuracy in Table 1 is not convincing. In [1], the PGD for epsilon=0.3 is 100% for non-robust models and around 55% for robust models, the number in the Table matches non of these.
>
> After reading your comment, we re-read the Madry et.al. (2017) for the PGD attacks, however there is a slight misinterpretation of the epsilon for the original paper and our reproduction.
> Kindly allow us to explain this misinterpretation.
> In the original paper, Madry et.al used epsilon=0.3 for the MNIST dataset while epsilon=8 was used for the CIFAR-10 dataset.
> Also, for the reasoning why we used a different parameter than the original paper is too keep all adversarial attacks comparable to each other.
> As epsilon is directly related to the degree of freedom an adversarial attack has for the perturbations.
> We kept the epsilon value as close as possible for all the different attacks we experimented with.
>
> Madry, A., Makelov, A., Schmidt, L., Tsipras, D., & Vladu, A. (2017). Towards deep learning models resistant to adversarial attacks. arXiv preprint arXiv:1706.06083.
>
> > 4. The writing of this paper is not clear and has gramma issues. For example, Table 1 and 2 miss information in Caption.
>
> Grammar issues corrected. Added more information in the caption of Table 1 and 2 (now merged in Table 1).
>
> > 5. The author should cite "Adversarial Examples Are a Natural Consequence of Test Error in Noise", which also studied the generalization and adversarial robustness.
>
> Reference added. We also discussed the reference paper briefly in Section 2 together with the added mathematical formulation.
>
> > 6. This evaluation measurement is not practical, it costs more computation than one evaluates a model directly using attacks. The measurement is also fragile under adversarial attacks, that one can feed in adversarial attacks to fool the score metrics, which is not convincing.
>
> We understand the concerns but the objective of this paper is not to propose a new/better cost effective evaluation measurement. The aim here is to understand the relation of robustness to the representation of models. We hope that the future architectures, training methodologies and algorithms would benefit from the analysis done here.

---

### Decision · Program_Chairs · 2019-12-19

**Decision:**

Reject

**Comment:**

The reviewers found the aim of the paper interesting (to connect representation quality with adversarial examples). However, the reviewers consistently pointed out writing issues, such as inaccurate or unsubstantiated claims, which are not appropriate for a scientific venue. The reviewers also found the experiments, which are on simple datasets, unconvincing.